# Hypothermia improves neuronal network recovery in a human-derived *in vitro* model of oxygen-deprivation

Eva J. H. F. Voogd [1]*, Marloes Thijs[1], Marloes R. Levers[1], Jeannette Hofmeijer[1,2‡], Monica Frega [1‡]*

1 Department of Clinical Neurophysiology, University of Twente, Enschede, The Netherlands, 2 Department of Neurology, Rijnstate Hospital, Arnhem, The Netherlands

‡ JH and MF are senior author on this work
* e.j.h.f.voogd@utwente.nl (EJHFV); m.frega@utwente.nl (MF)

**Data Availability Statement:** Data will be held in public repository mendeley titled: "Hypothermia improves neuronal network recovery in a human-

## Abstract

Mild therapeutic hypothermia showed potential neuroprotective properties during and after cerebral hypoxia or ischemia in experimental animal studies. However, in clinical trials, where hypothermia is mainly applied after reperfusion, results were divergent and neurophysiological effects unclear. In our current study, we employed human-derived neuronal networks to investigate how treatment with hypothermia during hypoxia influences neuronal functionality and whether it improves post-hypoxic recovery. We differentiated neuronal networks from human induced pluripotent stem cells on micro-electrode arrays (MEAs). We studied the effect of hypothermia (34°C)–as well hyperthermia (39°C) - on neuronal functionality during and after hypoxia using MEAs. We also studied the effects on the number of synaptic puncta and cell viability by immunocytochemistry. In comparison to neuronal networks under normothermia, we found that hypothermia during hypoxia improved functional neuronal network recovery, expressed as enhanced neuronal network activity. This was associated with prevention of synaptic loss during and after the hypoxic phase. Furthermore, hypothermia improved cell viability after the hypoxic phase. Instead, hyperthermia during hypoxia had detrimental effects, with an irreversible loss of neuronal network function, loss of synaptic puncta and decreased cell viability. Our results show potential neuroprotective properties of hypothermia occurring during hypoxia, indicating that administering hypothermia to bridge the time to reperfusion may be beneficial in clinical settings.

## Introduction

Mild therapeutic hypothermia has been the hallmark of treatment for patients with post-anoxic encephalopathy for about ten years. This therapeutic approach was based on results from two clinical trials conducted in 2002 in patients with transient cerebral hypoxia / ischemia after cardiac arrest, in which mild therapeutic hypothermia was associated with better neurological outcomes compared to maintaining patients in normothermic conditions [1,2].

derived in vitro model of oxygen-deprivation" Data is available at DOI:10.17632/3283rf85y6.1.

**Funding:** This research has been supported by an institutional research grant. The funders had no role in study design, data collection and analysis, decision to publish, or preparation of the manuscript.

**Competing interests:** The authors have declared that no competing interests exist.

The potential neuroprotective effect of hypothermia was further supported by studies in animal models of cerebral ischemia, showing that treatment at 32–35°C was associated with enhanced functional and histological recovery [3,4].

The effects of hypothermia in patients have become uncertain since 2013, when the large Targeted Temperature Management (TTM) trial showed no difference in outcome after cardiac arrest with treatment at 33°C and 36°C [5]. A systematic review conducted in 2021 also showed that patients treated with TTM 33°C had similar outcomes as those treated with 36°C [6]. The current general opinion is that the potential neuroprotective effects seen in previous clinical trials depended on the severity of hypoxia [7] and were likely driven by prevention of fever in the recovery phase, rather than to direct effects of hypothermia [8–11]. In patients with ischemic stroke, the potential neuroprotective effect of hypothermia remains even more unclear [12–15]. In all clinical studies, hypothermia was applied after hypoxia / ischemia, during the recovery phase.

The tentative mechanisms underpinning hypothermia's potential neuroprotective effects encompass a range of proposed actions, including reduction in metabolic demand [16–18], suppression of excitatory synaptic transmission [19–21], reduction of free radical production [22,23], anti-inflammatory effects [24–26], and attenuation of apoptosis [27–33]. Instead of targeting one specific step in the pathophysiological cascade, hypothermia is thought to modulate various pathomechanisms during and after cerebral ischemia [34].

The understanding of how hypothermia, in combination with ischemia, affect human neuronal function is incomplete. Hypothermia induced in living rats after cardiac arrest was found to reduce glutamate concentration [35]. Additionally, whole-cell patch clamp recordings of rat ventral horn neurons during ischemia and hypothermia indicated a decrease in the frequency of spontaneous excitatory post-synaptic currents as compared to normothermic conditions [21]. These findings have pointed towards attenuation of metabolic demand as a final common path to efficacy during hypoxia. Hyperthermia was found to elevate glutamate levels in the synaptic cleft [36]. However, a comprehensive understanding of hypothermia (and hyperthermia) effects during hypoxia on neuronal functionality and recovery is missing.

In the current study, we employed a human *in vitro* neuronal model previously described by us [37] to investigate how hypo- and hyperthermia influence the functioning of neuronal networks during hypoxia, as well as the effect of these temperature regimens on recovery after a hypoxia period. We found that hypothermia during hypoxia improved functional neuronal network recovery, which was associated with prevention of synaptic loss. In contrast, exposure to hyperthermia led to an irreversible loss of functionality in neuronal networks.

## Methods

This research was conducted in accordance with the principles embodied in the Declaration of Helsinki and local statutory requirements. The genetically modified organism (GMO) approval under which the cell lines have been used is IG22-071. All surgical and experimental procedures regarding animal primary cell lines followed Dutch and European laws and guidelines and were approved by the Centrale Commissie Dierproeven (CCD) (AVD11000202115663).

### Human iPSC differentiation towards neuronal networks

We used previously characterized mouse neuronal determinant neurogenin 2 (*Ngn2*) - and mouse neuronal determinant achuete-scute homolog 1 (*Ascl1)*-positive hiPSCs lines that were transfected, according to [38–40] and kindly provided by Prof. N. Nadif Kasri (Radboud University, Nijmegen, The Netherlands). All participants gave written informed consent to donate

their fibroblasts (Coriell Institute for medical research, GM25256 and KULSTEM iPSC core facility Leuven, Belgium, KSF-16-025).

We maintained *Ngn2-* and *Ascl1*-positive hiPSCs in E8 flex medium (Gibco, A2858501J) supplemented with RevitaCel (Thermo Fisher Scientific, A2644501), puromycin (0.5 µg/ml, Sigma Aldrich, P9620), and G418 (50 µg/ml, Sigma Aldrich, G8168) on 6-well plates pre-coated with Geltrex (Thermo Fisher Scientific A1413302), at 37°C and 5% $CO_2$. We refreshed the medium every 2–3 days, and passaged the cells twice per week using an enzyme-free reagent (ReLeSR, Stem Cell Technologies, 05872). Cells were checked for mycoplasma contamination.

We differentiated hiPSCs into glutamatergic cortical layer 2/3 neurons by overexpressing *Ngn2* (VectorBuilder, VB191004-1088vkr) with doxycycline (DOX, Sigma-Aldrich, D9891) treatment [39]. Next, we differentiated hiPSCs into GABAergic neurons by overexpressing *Ascl1* (Addgene 97329) with DOX treatment and supplementation with Forskolin (FSK, 10 µM, Sigma Aldrich, F6886) [40].

We used the following protocol to generate excitatory (*Ngn2*) and inhibitory (*Ascl1*) neuronal networks. On day in vitro 0 (DIV0), we co-plated *Ngn2*-positive and *Ascl1*-positive hiPSCs as single cells in 24-well micro-electrode arrays (MEAs) or on glass coverslips, pre-coated with poly-l-ornithine (50 µg/ml, Sigma Aldrich, P3655) and human laminin (20 µg/ml, Bioconnect LN521-05). We plated *Ngn2-* and *Ascl1*-positive cells in an 80:20 ratio (Fig 1A). The MEAs contain 24 independent wells with 12 embedded micro-electrodes (30 µm in diameter and spaced 200 µm apart) (Multi Channel Systems, Reutlingen, Germany).

To facilitate neuronal maturation, we added astrocytes obtained from cortices of newborn (P1) Wistar rats to hiPSC cultures in a 1:1 ratio on DIV2 [39]. On DIV3, we switched the medium to Neurobasal (Gibco, 10888022) and supplemented it with DOX (4 µg/ml), FSK (10 µM), B-27 (Gibco, 17504044), glutaMAX (2 mM, Thermo Scientific, 35050–038), primocin (0.1 µg/ml, Invivogen, ant-pm-2), neurotrophin-3 (10 ng/ml, Stemcell Technologies, 78074), and brain-derived neurotrophic factor (10 ng/ml, Stemcell Technologies, 78005). We added cytosine β-D-arabinofuranoside (2 µM, Sigma Aldrich, C1768) to eliminate proliferating cells.

Starting from this day, we changed half of the medium three times a week. From DIV10 onward, we added 2.5% fetal bovine serum (Sigma Aldrich, F7524) to support astrocyte viability. After DIV14, we removed DOX and FSK. We maintained the cells were in an incubator at 37°C with 80% humidity and 5% $CO_2$ until the experiment on DIV49.

We applied stringent inclusion criteria for all neuronal networks [41]. We included neuronal networks when they showed good quality (i.e. cell density allowing for proper neuron-electrode coupling and even distribution of cells).

## Experimental protocol

We subjected 24-well MEAs and glass coverslips to the following experimental protocols. We placed neuronal networks on MEA or glass coverslips in climate-controlled chambers allowing us to establish normoxia (i.e., normoxic neuronal networks; 20% $O_2$ / 75% $N_2$ / 5% $CO_2$) or hypoxia (i.e., hypoxic neuronal networks; 2% $O_2$ / 93% $N_2$ / 5% $CO_2$) (Fig 1B) [42]. We investigated three different temperature regimens in normoxic and hypoxic neuronal networks: 34°C (hypothermia), 37°C (normothermia), and 39°C (hyperthermia).

## Electrophysiological outcome measures

Before each experiment, we shielded the 24-well MEA with a Breath Easier sealing membrane (Sigma Aldrich, Z763624). This membrane facilitated gas exchange. Spontaneous electrophysiological activity in hiPSC-derived neuronal networks was recorded as follows: a 10-minute

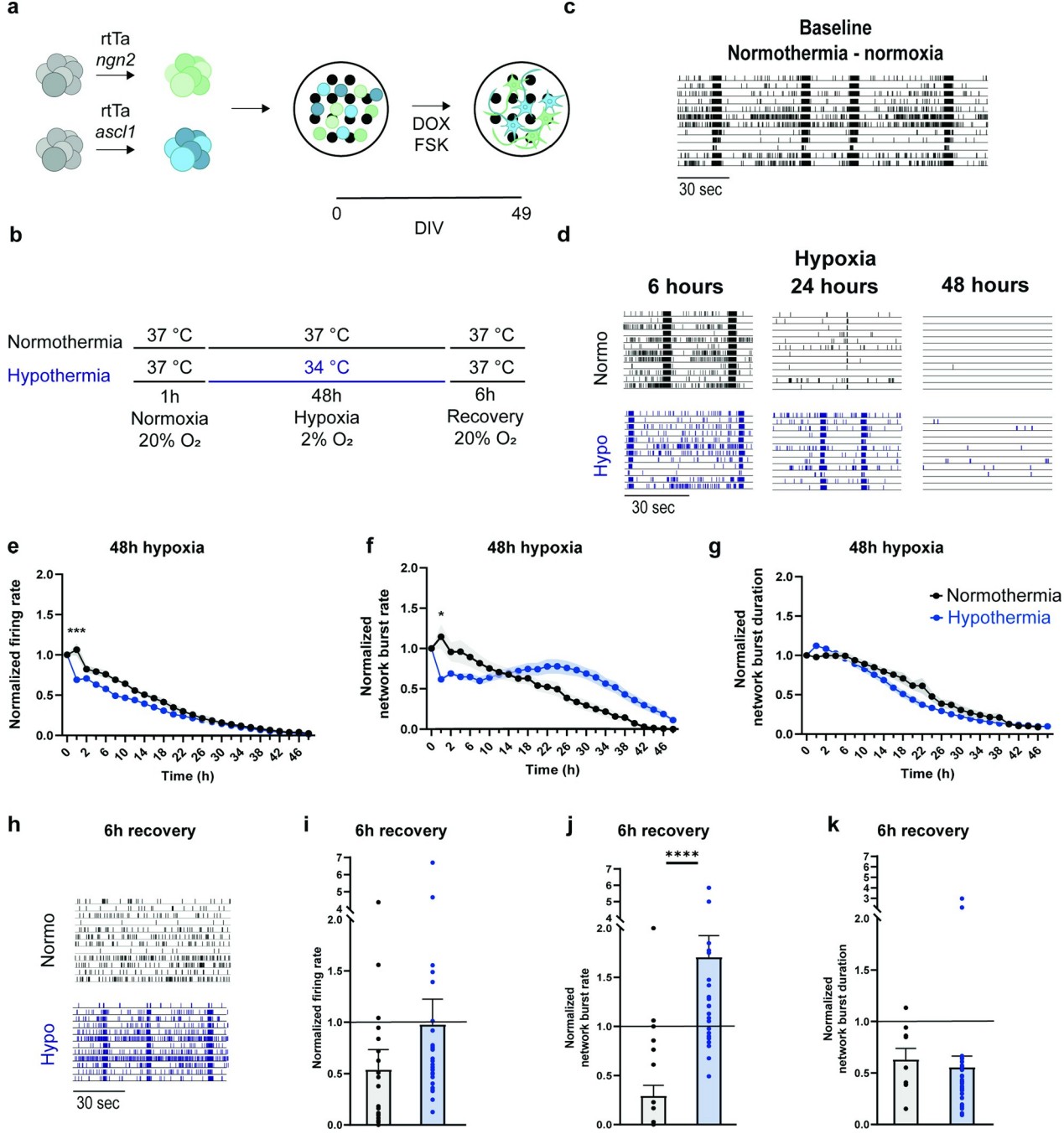

**Fig 1. Hypothermia improves functional recovery in hypoxic neuronal networks. a.** Schematic presentation of the differentiation protocol of human induced pluripotent stem cells (hiPSCs) into excitatory and inhibitory neurons on micro-electrode arrays (MEAs). **b.** Timeline of the experimental protocol. After initial exposure to normoxia (20% $O_2$) and normothermia (37˚C), neuronal networks are exposed to hypoxia (2% $O_2$) for 48 hours. Afterwards, neuronal networks are exposed to 6 hours of normoxia (recovery). During the hypoxia period temperatures were set to 37˚C (normothermia) or 34˚C (hypothermia). **c-d.** Representative raster plot showing **c.** three minutes of electrophysiological activity during baseline (normothermia and normoxia and **d.** one minute of electrophysiological activity at normothermia (black) and hypothermia (blue) at different time points during hypoxia (6 hours, 24 hours and 48 hours after the onset of hypoxia). **e-g.** Graphs showing the effect of 48 hours of hypoxia combined with normothermia (black; n = 23) or hypothermia (blue; n = 30) recorded for 10 minutes every two hours on the **e.** firing rate, **f.** network burst rate and **g.** network burst duration. **h.** Representative raster plots showing one minute of electrophysiological activity exhibited by neuronal networks exposed at normothermia (black) and hypothermia (blue) during hypoxia recorded 6 hours after normothermia and normoxia have been re-established (recovery). **i-k.** Bar graphs showing the effect after 6 of hours recovery (normoxia and normothermia) in neuronal networks exposed to normothermia (black; n = 23) or hypothermia (blue; n = 30) during hypoxia recorded for 10 minutes on **i.** firing rate, **j.** network burst rate and **k.** network burst duration. Black lines across bars represent normalized baseline. *P < 0.05, **P < 0.005, ***P < 0.0005, ****P < 0.0001.

Two-way ANOVA with Tukey's multiple comparison analysis were performed between conditions. P < 0.0001 are not shown after the first time this value was found. Exact p values are reported in S1 Table.

baseline during normoxia and normothermia (after 30 minutes of stabilization). It took 1 hour to reach the desired oxygen level in the neuronal networks [37]. Then 10-minute recordings every 2 hours during 48 hours of hypoxia with temperature treatment were conducted. Following the hypoxia phase, we re-established normoxia and normothermia to assess recovery, with a 10-minute recording after 6 hours.

We utilized the Multiwell-MEA system (Multi Channel Systems, Reutlingen, Germany) to assess electrophysiological activity and analyzed the activity through the Multiwell-Screen software. Signals were sampled at 10 kHz frequency and filtered via a high-pass filter (2nd order Butterworth filter, 100 Hz) and a low-pass filter (4th order Butterworth filter, 3.5 kHz).

We conducted data analysis using a combination of the Multiwell Analyzer software (Multi Channel Systems, Reutlingen, Germany) and custom MATLAB scripts (The MathWorks, Natick, MA, USA).

**Spike detection.** We identified spikes when the voltage surpassed 5 times the standard deviation of the baseline noise. We deemed an electrode active if it displayed a minimum of 0.1 spikes per second. Subsequently, we computed the mean firing rate (MFR) by determining the spike count per electrode over time and averaging across all active electrodes on the MEA.

**Burst detection.** We identified single-channel bursts as sequences of spikes containing a minimum of 4 spikes, each with a maximum inter-spike interval of 50 ms. A minimum interval of 100 ms was used to differentiate between consecutive bursts. A channel was categorized as a bursting channel if it displayed a frequency of at least 0.4 bursts per minute.

**Network burst detection.** We defined a network burst as a sequence of temporally overlapping bursts from various channels. For a sequence to be detected as a network burst, it required a minimum of six distinct bursting channels, with at least four channels displaying concurrent bursting activity at some point during the sequence. Following detection, parameters characterizing the network burst were extracted: network burst rate (NBR, the number of network bursts identified over time) and network burst duration (NBD, the average duration of all detected network bursts).

We deemed neuronal networks active and included in analysis, when neuronal networks displayed an MFR > 0.1 spike per second and NBR > 1 network burst per minute.

## Immunocytochemical outcome measures

To investigate the effects of the different temperature regimens on cell viability, we applied immunocytochemical staining at normoxia, 24 and 48 hours of hypoxia and 6 and 24 hours of recovery. To investigate the effects of different temperatures on synaptic puncta, we applied immunocytochemical staining at normoxia, 6, 24 and 48 hours of hypoxia and 6 and 24 hours of recovery. To investigate the effects of different temperature regimens on cell viability, we applied immunocytochemical staining at normoxia, 24 and 48 hours of hypoxia and 6 and 24 hours of recovery.

**Cell viability including apoptosis.** We added Cell Event caspase3/7 (1:1000; Thermo-Scientific) to the cells to stain for apoptosis and incubated them for 30 min at 37°C. After incubation, we initiated the hypoxic period for 24, and 48 hours in combination with the different temperature regimens. For neuronal networks subjected to recovery we initiated the hypoxic period for 48 hours followed by recovery of 6 or 24 hours. At the end of the hypoxic period, we added propidium iodide (PI, 1:500; Invitrogen) for 15 min at room temperature (RT) to stain the dead cells. We then washed the cells with phosphate-buffered saline (PBS) and fixed them

with 4% paraformaldehyde (Sigma Aldrich) for 15 min at RT. Finally, we added DAPI (1:1000; Sigma Aldrich) for 20 min at RT, washed the cells with PBS, and mounted them with mowiol (Sigma Aldrich).

We captured images of ten random fields at 40× magnification using a Nikon Eclipse 50i epi-fluorescence microscope. To investigate cell viability, we conducted image analysis using custom MATLAB scripts to preprocess the captured images (Fig 3A and 3D). Subsequently, a blind manual cell count was performed, distinguishing cells which were considered live when only DAPI (blue) was visible, apoptotic when positive for DAPI (blue) and Cell Event (green) and dead when positive for DAPI (blue), Cell Event (green) and PI (red).

**Synaptic puncta.** We stained neuronal networks for synapsin1/2 with the following protocol. HiPSC-derived neuronal networks that had been cultured on glass coverslips and subjected to experimental treatment (i.e. hypoxia in combination with one of the temperature regimens), were fixed using 3.7% paraformaldehyde (PFA, Sigma Aldrich, F8775) for 15 minutes at room temperature (RT), then rinsed with Dulbecco's phosphate-buffered saline (dPBS, Gibco, 14190–169) and stored in dPBS at 4˚C until staining. To begin, we permeabilized samples using 0.2% triton X-100 (Sigma Aldrich, 93443) in dPBS and subsequently washed with dPBS. Following this, We added a blocking buffer (2% bovine serum albumin, Sigma Aldrich, A1595, in dPBS) for 30 minutes at RT to inhibit non-specific binding.

To identify synaptic puncta, we stained the neuronal networks with mouse anti-MAP2 (1:1000, Sigma Aldrich, M4403-50) and guinea pig anti-synapsin1/2 (1:1000, Synaptic Systems, 106004) diluted in blocking buffer and incubated overnight at 4˚C. Next, we applied secondary antibodies for 1 hour at RT, including goat anti-mouse (1:2000, Invitrogen, AF488, A11029) and goat anti-guinea pig (1:2000, Invitrogen, AF647, A21450). Subsequently, we stained the nuclei with DAPI (1:1000, Sigma Aldrich, D9542) for 10 minutes at RT. After another round of careful washing, we mounted the samples were using Mowiol, dried, and stored at 4˚C.

We captured images of ten random fields at a 60x magnification using a Nikon Eclipse 50i Epi-Fluorescence microscope (Nikon, Japan). We quantified the number of synaptic puncta by employing SynD scripts from Schmitz, Hjorth, et al. 2011 [43] using MATLAB and expressed as the number of synapsin1/2 positive puncta per 10 μm of MAP2 positive neurite.

## Statistical analysis

We conducted statistical analysis using GraphPad Prism 9 (GraphPad Software, Inc., CA, USA). Prior to analysis, we verified normal distribution through the Kolmogorov–Smirnov normality test.

To enable effective comparisons, we normalized all parameter values across different experiment phases relative to their baseline values. We employed two-way Analysis of Variance (ANOVA) with multiple comparisons tests to test differences between the various experimental conditions. In each case, comparisons were made against the corresponding normalized baseline. For different temperature groups, comparisons were drawn against the normothermic neuronal networks.

For the immunocytochemical readouts, comparisons were made within each specific time point of fixation, as each neuronal network is an independent sample. This approach ensures that differences observed between experimental conditions are not influenced by variations in fixation times across networks, maintaining the integrity of the results for each condition.

We performed identification of outliers by employing Robust Regression and Outlier removal (ROUT) method with Q = 1%, meaning that data points that fell in the lower and upper 1% of the normal distribution were considered potential outliers. We presented our data as mean ± standard error of the mean (SEM). The figure legends provide information

about the number of independent neuronal networks included in each experiment. Elaborate statistical details are provided in S1–S5 Tables. P < 0.05 was considered statistically significant.

## Results

### Hypoxia affects neuronal network functioning

After seven weeks *in vitro*, hiPSC-derived neurons grown on MEAs under normothermic and normoxic conditions established functional networks exhibiting spikes, bursts and network bursts (Fig 1C, "Baseline"). When subjected to hypoxia (Fig 1B–1G, "Normothermia"), these networks experienced a gradual decline in both firing (MFR) and synchronous activity (NBR), leading to a complete loss of activity within 42 hours. Despite subsequent reoxygenation, recovery remained incomplete, with a restoration of 54% in MFR, 29% in NBR, and 63% in NBD as compared to baseline (Fig 1H–1K). We returned the neuronal networks to physiological conditions (i.e., normoxia and normothermia) and recorded activity 6 hours later to investigate recovery. We observed that the MFR recovered to 54% of baseline level, the NBR returned to 29% of baseline level and the NBD returned to 63% of baseline level.

### Hypothermia improves functional recovery in hypoxic neuronal networks

To investigate how hypothermia (34˚C) influences neuronal network functioning during and after the hypoxia period, we exposed neuronal networks to hypothermia throughout hypoxia and recorded baseline activity and subsequently after the induction of hypoxia we recorded 10 minutes of activity every 2 hours during 48 hours (Fig 1B, "Hypothermia"). Under hypothermia, neuronal networks displayed a similar gradual decrease in MFR and NBD during hypoxia as those under normothermic conditions (Fig 1E and 1G). However, NBR did not follow a gradual trend as under normothermia: it initially dropped to about 50% of the baseline (p = 0.0425) where it remained stable for the first 34 hours of hypoxia. Thereafter, it gradually decreased until 48 hours, when all synchronous events ceased (Fig 1F).

During the recovery phase we observed that neuronal networks exposed to hypothermia showed higher activity as compared to neuronal networks kept under normothermia (Fig 1H–1K). In particular, hypothermia led to a 1.7-fold increase of NBR as compared to baseline (p < 0.0001).

### Hyperthermia impedes functional recovery in hypoxic neuronal networks

Our subsequent investigation aimed to elucidate how hyperthermia (39˚C) impacts the functionality and recovery of hypoxic neuronal networks (Fig 2A, "Hyperthermia"). Hyperthermia exposure resulted in elevated MFR values within the first 14 hours of hypoxia, followed by a gradual decline that was similar as in neuronal networks under normothermic conditions (Fig 2B and 2C; p < 0.05). Network burst activity also evolved differently under hyperthermic as compared to normothermic conditions (Fig 2B and 2D). Instead of a gradual decrease in NBR, networks exposed to hyperthermia showed an initial rise, resulting in a 1.5-fold increase as compared to baseline levels. This elevated level of synchronous activity was maintained for the first 24 hours of hypoxia. Afterwards, we observed a prompt decrease in the appearance of network bursts, that entirely disappeared by 42 hours of hypoxia. This timing of NBR loss in hyperthermic networks mirrored the one observed in normothermic networks. NBD gradually decreased over the 48-hour hypoxia period under hyperthermic conditions, essentially similar as in neuronal networks under normothermic conditions (Fig 2E). Upon returning to physiological conditions for 6 hours, hypoxic neuronal networks exposed to hyperthermia showed

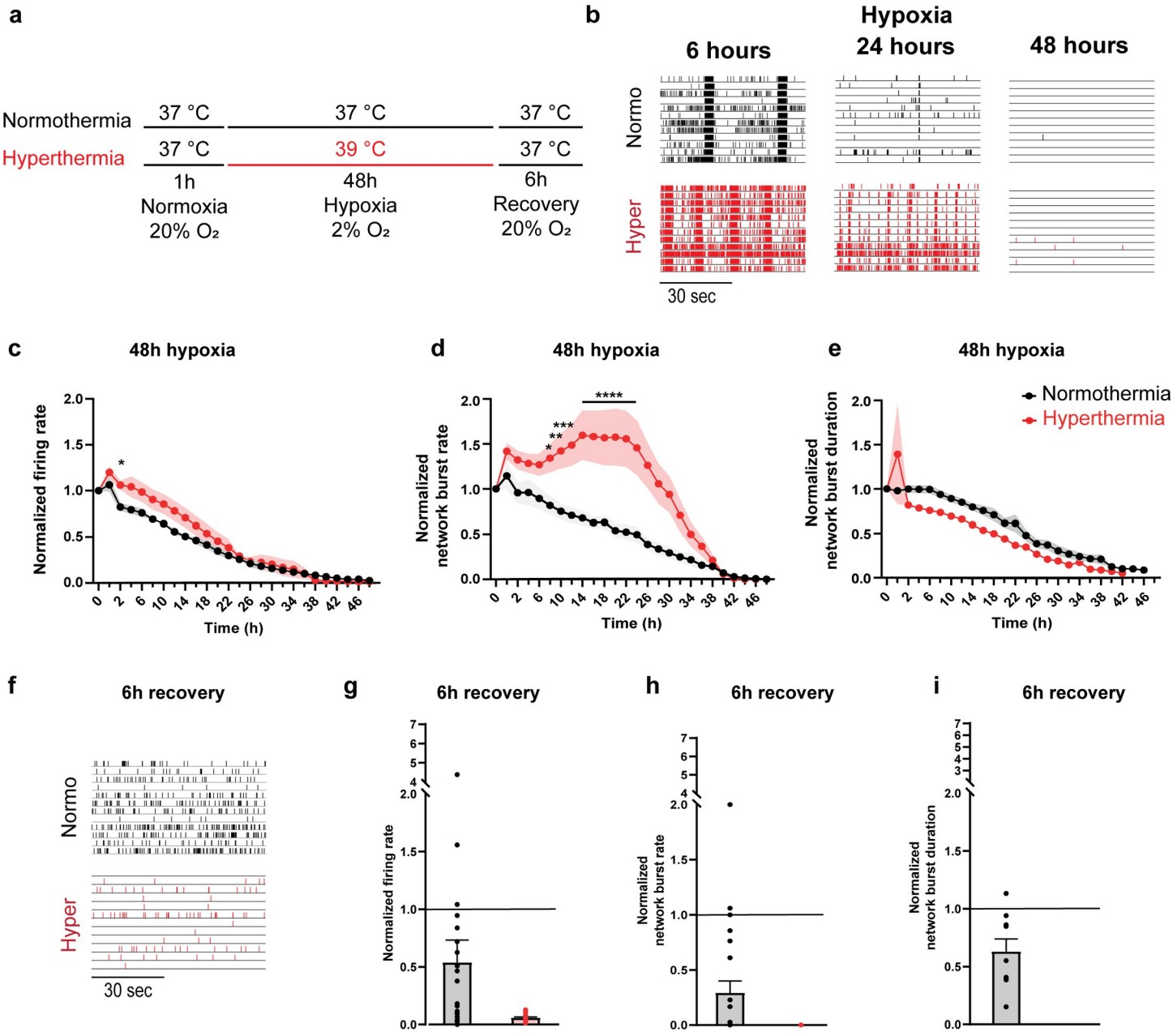

**Fig 2. Hyperthermia impedes functional recovery in hypoxic neuronal networks. a.** Timeline of the experimental protocol. After initial exposure to normoxia (20% $O_2$) and normothermia (37˚C), neuronal networks are exposed to hypoxia (2% $O_2$) for 48 hours. Afterwards, neuronal networks are exposed to 6 hours of normoxia (recovery). During the hypoxia period temperatures were set to 37˚C (normothermia) or 39˚C (hyperthermia). **b.** Representative raster plots showing one minute of electrophysiological activity at normothermia (black) and hyperthermia (red) at different time points during hypoxia (6 hours, 24 hours and 48 hours). **c-e.** Graphs showing the effect of 48 hours of hypoxia combined with normothermia (black; n = 23) or hyperthermia (red; n = 42) recorded for 10 minutes every two hours on the **c.** firing rate, **d.** network burst rate and **e.** network burst duration. **f.** Representative raster plots showing one minute of electrophysiological activity exhibited by neuronal networks exposed to normothermia (black) and hyperthermia (blue) during hypoxia recorded 6 hours after normothermia and normoxia have been re-established (recovery). **g-i.** Bar graphs showing the effect after 6 hours of recovery (normoxia and normothermia) in neuronal networks exposed to normothermia (black; n = 23) or hyperthermia (red; n = 42) recorded for 10 minutes on **g.** firing rate, **h.** network burst rate and **i.** network burst duration. Black lines across bar graphs represent normalized baseline. *P < 0.05, **P < 0.005, ***P < 0.0005, ****P < 0.0001. Two-way ANOVA with Tukey's multiple comparison analyses were performed between conditions. P < 0.0001 are not shown after the first time this value was found. Exact p values are reported in S2 Table.

stronger functional impairments in comparison to the ones exposed to normothermia. In particular, we observed a minimal firing and absent synchronous activity as compared to baseline (p < 0.0001) (Fig 1F–1I).

## Hypothermia improves cell viability after hypoxia

We next investigated the effect of normo-, hypo-, and hyperthermia on cell viability (Fig 3A and 3B). After 24 hours of hypoxia, under normothermia we observed 36% live cells, 11% apoptotic cells, and 52% dead cells. Under hypothermia, there was a slightly higher percentage of

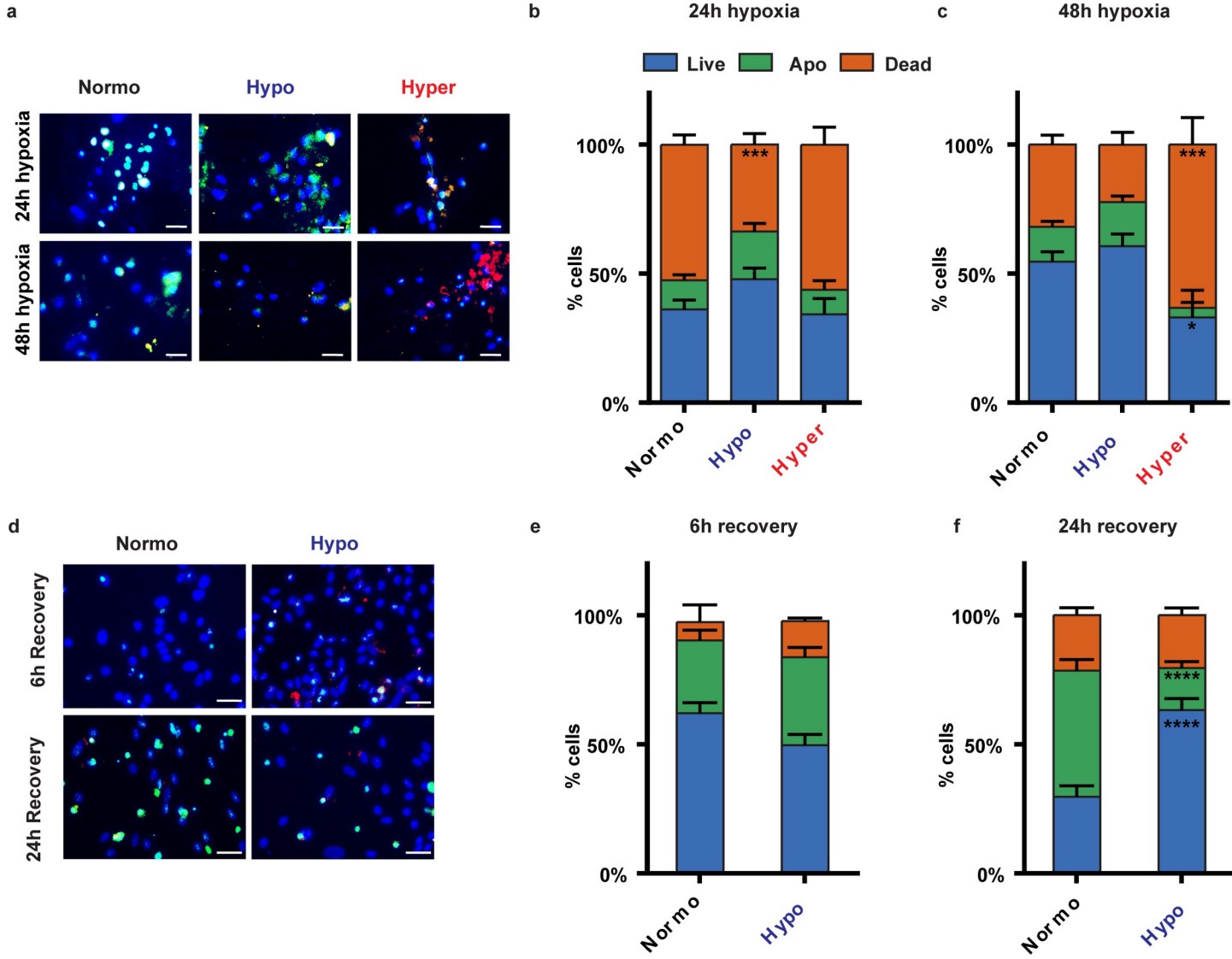

**Fig 3. Hypothermia improves cell viability after hypoxia. a.** Representative images showing cell viability including apoptosis of neuronal networks exposed to 24 or 48 hours hypoxia and normothermia, hypothermia or hyperthermia. Neuronal networks stained for DAPI (blue), Cell event Caspase-3/7 (green) and propidium iodine (red). Scale bar 30 μm. **b-c.** Stacked bar graphs showing the percentage of live, apoptotic (apo) and dead cells after **b.** 24 hours of hypoxia and **c.** 48 hours of hypoxia for normothermia (n = 9), hypothermia (n = 7) and hyperthermia (n = 3). **d.** Representative images showing cell viability including apoptosis of neuronal networks exposed to 6 or 24 hours of recovery after hypoxia and normothermia or hypothermia. Neuronal networks were stained for DAPI (blue), Cell event Caspase-3/7 (green) and propidium iodine (red). **e-f.** Stacked bar graphs showing the percentage of live, apoptotic (apo) and dead cells after **e.** 6 hours of recovery or **f.** 24 hours of recovery for normothermia (n = 4) and hypothermia (n = 6). *P < 0.05, **P < 0.005, ***P < 0.0005, ****P < 0.0001. Two-way ANOVA with **b-c.** Tukey's multiple comparisons and e-f. Sidak's multiple comparisons test. Exact p values are reported in S3 Table.

live cells (48%) and apoptotic cells (19%), with significantly fewer dead cells (34%) compared to normothermia (p = 0.0006). Hyperthermia resulted in 34% live cells, 10% apoptotic cells, and 56% dead cells, showing a similar pattern to normothermia with no significant differences in live, apoptotic or dead cell counts.

After 48 hours of hypoxia, hypothermic networks maintained a higher percentage of live cells (61%) compared to normothermic networks (55%), and fewer dead cells (22% vs. 32%), although this difference was not statistically significant. Hyperthermia was associated with a significant decline in cell viability, exhibiting the lowest percentage of live cells (33%) and the highest percentage of dead cells (63%). Significant differences were found between normothermic and hyperthermic networks in both live cell count (p = 0.0129) and dead cell count (p = 0.0001).

Next, we examined the recovery of neuronal networks under normothermic and hypothermic conditions, excluding hyperthermic networks because of the absence of functional recovery after hypoxia (Fig 3D–3F). After 6 hours of recovery, under normothermia neuronal networks showed a higher percentage of live cells (62%) compared to hypothermic networks, although not statistically significant. However, after 24 hours of recovery, hypothermic neuronal networks demonstrated significantly better outcomes, with 93% live cells compared to 37% in normothermic networks (p < 0.0001). Additionally, hypothermic networks had significantly fewer apoptotic cells (16%) compared to normothermic networks (49%; p < 0.0001).

## Hypothermia prevents loss of synaptic puncta during and after hypoxia

Next, we investigated whether the divergent activity patterns observed in hypoxic neuronal networks exposed to normo-, hypo- and hyperthermia were associated with changes in the number of synaptic puncta (Fig 4A–4D).

We found that the decrease of activity observed in hypoxic neuronal networks was associated with a reduction in synaptic puncta. Under normothermia, we found a 32% loss at 6 hours (p = 0.0462), a 20% loss at 24 hours (p = 0.787), and a 50% loss at 48 hours (p < 0.0001) of hypoxia when compared to baseline.

Instead, under hypothermia, we observed a significant increase in synaptic puncta during hypoxia. There was a significantly higher number of synaptic puncta after 6, 24, and 48 hours of hypoxia as compared with during normothermia (p < 0.0001), and the number of synaptic puncta was also statistically higher as compared to baseline (p < 0,0001).

Finally, hyperthermia was associated with a similar loss of synaptic puncta as normothermia after 6, 24, and 48 hours of hypoxia (p = 0,9254, p = 0,1738, p > 0.9999, respectively).

Next, we examined the recovery of neuronal networks under normothermic and hypothermic conditions, excluding hyperthermic networks because of the absence of functional recovery after hypoxia (Fig 4E–4G). After 6 hours of recovery, normothermic and hypothermic neuronal networks had 0.84 ± 0.07 synaptic puncta per 10 μm and 0.99 ± 0.08 synaptic puncta per 10 μm, respectively (p > 0.9999). After 24 hours of recovery, hypothermic neuronal networks had a higher number of synaptic puncta per 10 μm than normothermic neuronal networks (1.6 ± 0.11 vs. 0.8 ± 0.09; p<0.0001).

## Effect of hypo- and hyperthermia on neuronal network functioning in normoxia

Subsequently, we studied the effect of hypo- and hyperthermia on functional activity of normoxic neuronal networks (Fig 5A).

Normoxic neuronal networks exposed to hypothermia showed a gradual decline in MFR, NBR, and NBD over the course of 48 hours (Fig 5B–5E). When physiological conditions were

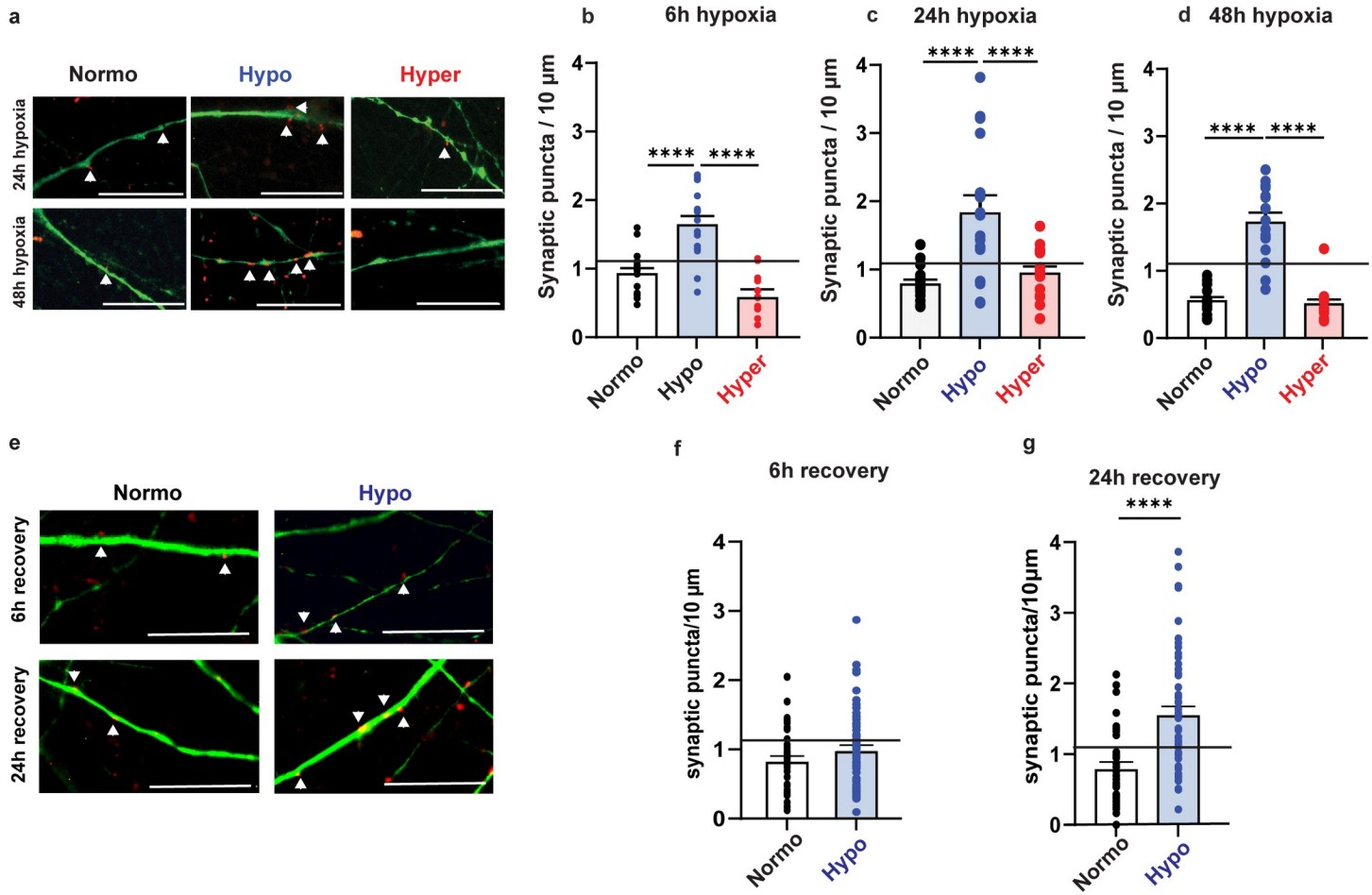

**Fig 4. Number of synaptic puncta increased during and after hypoxia in hypothermic neuronal networks. a.** Representative images showing synaptic puncta (arrowheads, positive for synapsin 1/2; red) along the dendrite (positive for MAP2; green; nuclei positive for DAPI; blue) in neuronal networks after 24 hours and 48 hours of hypoxia during normothermia (37˚C), hypothermia (34˚C) and hyperthermia (39˚C). Scale bar: 10 μm. **b-d.** Bar graphs showing the number of synaptic puncta per 10 μm dendrite in neurons exposed to hypoxia combined with normothermia (black; n = 6), hypothermia (blue; n = 2) or hyperthermia (red; n = 4) after **b.** 6 hours, **c.** 24 hours and **d.** 48 hours. **e.** Representative images showing synaptic puncta (arrowheads, positive for synapsin 1/2; red) along the dendrite (positive for MAP2; green; nuclei positive for DAPI; blue) in normothermic and hypothermic neuronal networks after 6 and 24 hours of recovery. Scale bar: 10 μm. **f-g.** Bar graphs showing the number of synaptic puncta per 10 μm dendrite in neurons exposed to normothermia (n = 4) or hypothermia (n = 6) after **f.** 6 hours and **g.** 24 hours of recovery. Black line represents baseline number of synaptic puncta (n = 11). *P < 0.05, **P < 0.005, ***P < 0.0005, ****P < 0.0001. Two-way ANOVA with Tukey's multiple comparison analyses were performed between conditions. Exact p values are reported in S4 Table.

re-established, neuronal network functioning of normoxic cultures was impaired (Fig 5I–5K). In particular, we observed that MFR reached only 31% of the baseline activity (p <0.0001) and network bursts remained absent (p < 0.0001).

Neuronal networks exposed to hyperthermia under normoxic conditions showed a gradual decline in MFR, NBR and NBD (Fig 5F–5H). When physiological conditions were re-established, we observed that MFR reached 28% of the baseline activity (p < 0.0001), while NBR reached only 7% of baseline activity (p < 0.0001) (Fig 5K–5M).

## Discussion

In this study, we used a validated human in vitro model to investigate the vulnerability of neuronal networks to hypoxia [37]. In addition to the human nature, advantages of the model include precise control over environmental conditions, such as oxygen levels and

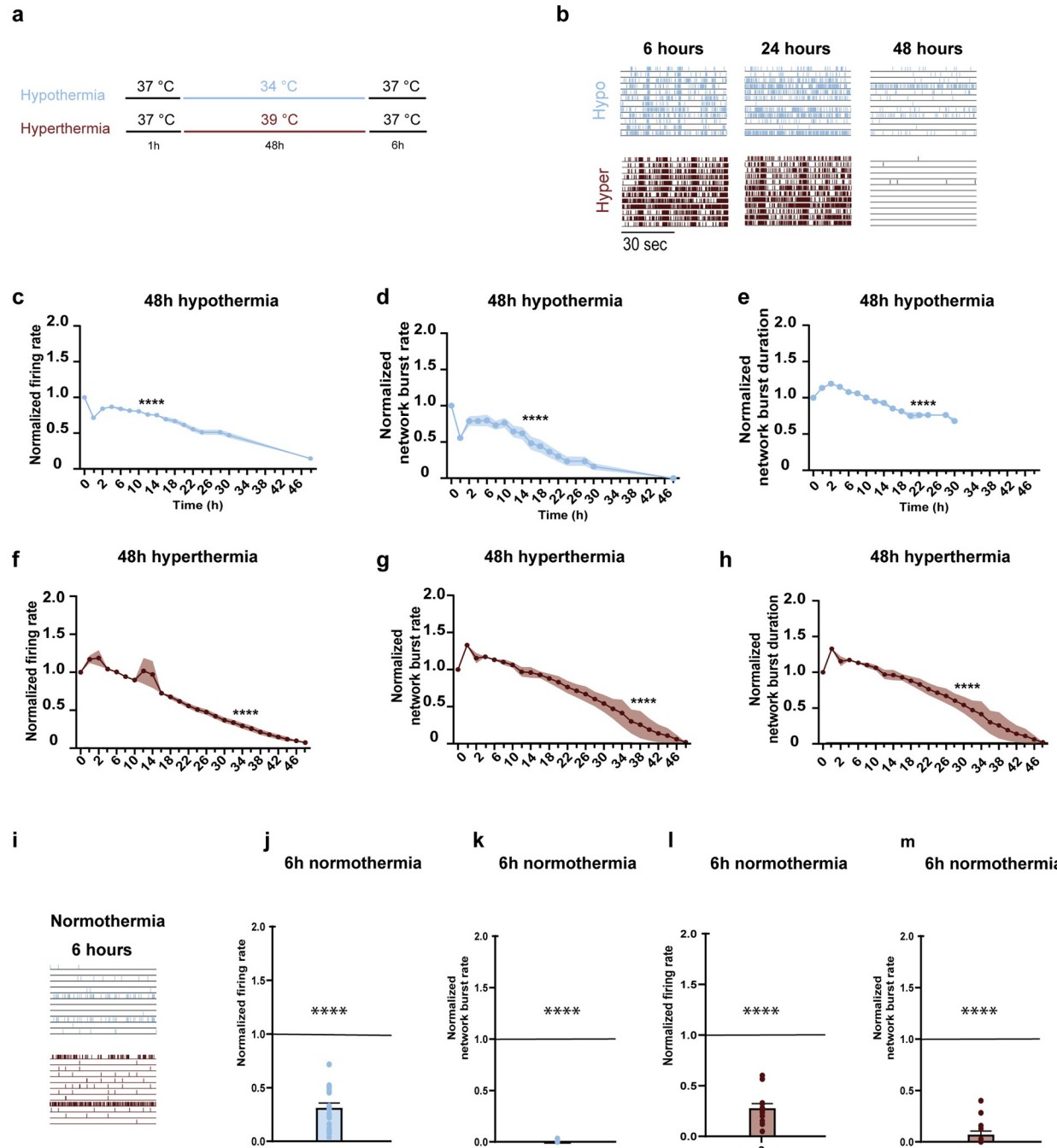

**Fig 5. Hypothermia and hyperthermia impair neuronal functioning in normoxia. a.** Timeline of the experimental protocol. After initial exposure to normoxia (20% $O_2$) and normothermia (37°C), neuronal networks are exposed to hypothermia (34°C) or hyperthermia (39°C) for 48 hours. Afterwards, neuronal networks are exposed to 6 hours of normoxia and normothermia. **b.** Representative raster plots showing one minute of electrophysiological activity exhibited by neuronal networks exposed to hypothermia or hyperthermia during normoxia at different time points (6 hours, 24 hours, 48 hours and 6 hours after re-establishing normothermia). **c-e.** Graphs showing the effect of 48 hours of hypothermia combined with normoxia (n = 19) recorded for 10 minutes every two hours on the **c.** firing rate, **d.** network burst rate and **e.** network burst duration. **f-h.** Graphs showing the effect of 48 hours of hyperthermia combined with normoxia (n = 14) on the **f.** firing rate, **g.** network burst rate and **h.** network burst duration. **i-j.** Bar graphs showing the effect of 6 hours recovery in neuronal networks exposed to hypothermia during normoxia (n = 19) recorded for 10 minutes on **i.** firing rate and **j.** network burst rate. **k-l.** Bar graphs showing the effect of 6 hours after re-establishing normothermia in neuronal networks exposed to hyperthermia during normoxia (n = 14) recorded for 10 minutes on **k.** firing rate and **l.** network burst rate. Black lines across bar graphs represent baseline. One-way ANOVA with Dunnet's multiple comparison test was performed for **c-h.** P < 0.0001 are not shown after the first time this value was found. Exact p values are reported in S5 Table. Two-way ANOVA with Sidak's multiple comparisons test were performed for **j -k.**

temperatures, providing a clear understanding of how these variables influence neuronal network functions, synaptic integrity, cell viability and recovery potential.

Our findings show that hypothermia (34˚C) during hypoxia stabilized NBR and slowed the decline. The 1.7-fold increase in NBR compared to baseline suggests enhanced recovery of neuronal networks under hypothermic conditions. This result was further supported by an increased number of synaptic puncta in neuronal networks subjected to hypothermia during and after hypoxia, as well was improved cell viability after hypoxia. In contrast, hyperthermia (39˚C) initially elevated NBR, followed by a rapid decline. After 6 hours of reoxygenation, networks exposed to hyperthermia exhibited pronounced impairments, with minimal MFR and absent NBR. These networks also showed a persistent loss of synaptic puncta and cell viability. These results suggest that hypothermia may protect neuronal networks during hypoxia and improve recovery, whereas hyperthermia exacerbates functional impairments.

## Hypoxia is deleterious for network functioning and cell viability in neuronal networks

Our results showed that hypoxia led to a persistent decrease in neuronal functionality, synaptic puncta and cell viability. Furthermore, subsequent re-oxygenation led to partial recovery in these readouts. This was in line with earlier studies [37,44]. After the hypoxic period, we found that the number of synaptic puncta was reduced by half. Synaptic failure leading to a reduction of functional synapses, and ultimately loss of synapses, is a known phenomenon of hypoxic injury in neurons [42,45–49]. This effect of hypoxia on neurons leads to compromised neuronal function [50], eventually leading to decreased viability [37]. The remaining synapses may still contribute to plasticity, offering potential for recovery once the hypoxic stress subsides. Additionally, while our study focused on overall synaptic puncta, distinguishing between excitatory and inhibitory synapses could provide further insight into cell-type-specific vulnerabilities. A prior study has shown that inhibitory neurons are more vulnerable to hypoxic stress, leading to an increased excitation/inhibition ratio [51].

## Hypothermia enhances functional recovery in hypoxic neuronal networks

We demonstrated that hypothermia during hypoxia is potentially neuroprotective. This is distinct from previous clinical trials, where hypothermia was studied after cardiac arrest or ischemic stroke, in the recovery phase[24–26,52–56].

We showed that the neuroprotective effect of hypothermia during hypoxia resulted in elevated NBR after re-establishment of physiological conditions. Additionally, we observed an increased number of synaptic puncta during and after hypoxia, as well as higher cell viability after hypoxia. The increase in synaptic puncta suggests that hypothermia may promote synaptic homeostasis, preserving or restoring synaptic connections critical for maintaining neuronal communication during metabolic stress. This compensatory mechanism likely supports neuronal survival and functional recovery [57]. The correlation between these aspects suggests that the preservation of synaptic puncta and improved cell viability might contribute to the observed improvements in functional recovery. Our findings are in line with previous studies. Hypothermia during hypoxia enhanced electrophysiological recovery in rat hippocampal slices [58]. Furthermore, hypoxia-induced synaptic disfunction was effectively blocked by hypothermia in rats [59]. Additionally, previous findings indicating improved synaptic plasticity with hypothermia following ischemic stroke further support the hypothesis that the increased number of synaptic puncta during hypothermia in hypoxia contributes to the surge in activity after physiological conditions were re-established [60]. Another study demonstrated that hypothermia during asphyxia can prevent neuronal death by impeding actin cytoskeleton changes in

dendritic spines [61]. Neuronal cell death is also prevented by hypothermia in a model of oxygen and glucose deprivation (OGD) [62].

## Hypothermia in normoxia is deleterious for neuronal network functioning

Hypothermia in normoxia led to a similar gradual reduction of neuronal network activity, but this time without return to baseline. This aligns with prior research on MEA-cultivated dissociated neuronal networks, where a 20-hour hypothermic exposure (19°C) led to decreased network burst rate and spontaneous firing [63]. However, while in their work Rubinsky and coworkers observed a return to baseline levels after re-establishment of normothermia [63], neuronal activity in our *in vitro* model remained low. This is likely because of the different durations of hypothermia exposure utilized in the two studies (i.e., 20 vs. 48 hours). In fact, it is recognized that prolonged reduction or suppression of activity is detrimental to neuronal networks [37,64]. It has been shown that hypothermia during normoxia reinforces inhibitory synapses in hippocampal neurons [65]. Furthermore, hypothermia exerts a modulatory influence on neuronal function by affecting the behavior of voltage-gated membrane channels (i.e., potassium and sodium), leading to a decrease in neuronal firing [66–68]. Thus, the divergent effects of hypothermia in normoxic and hypoxic neuronal networks suggest that neuroprotection through hypothermia targets pathomechanisms that specifically occur under hypoxia.

## Hyperthermia impedes functional recovery in hypoxic neuronal networks

In contrast to hypothermia, hyperthermia led to detrimental effects in hypoxic neuronal networks. This is in line with what is observed in patient populations of ischemic stroke, where hyperthermia (i.e., fever) was found to increase death, disability and discharge to hospice [8,9,69,70]. Previous preclinical studies have shown that, under hypoxia, hyperthermia exacerbated electrophysiological changes and elevated intracellular $Ca^{2+}$ levels, leading to a lack of recovery [58]. Further, it increased caspase-3 activity and thus apoptosis [71], similarly to what we found. Hyperthermia induced elevated glutamate levels in the synaptic cleft, probably resulting in increased excitability [36]. Additionally, hypoxia influenced inhibitory synapses, contributing to heightened excitability [72]. This increased excitability aligns with the observed initial rise in NBR in neuronal networks under hyperthermia.

## Hyperthermia in normoxia is deleterious for neuronal network activity

However, while in hypoxic neuronal networks NBR remains elevated for prolonged time, in normoxic conditions the initial rise was rapidly followed by a gradual decline. This is in line with previous work showing that hyperthermia in normoxia correlated with decreased network bursts and firing rates in rodent neuronal networks on MEA [73]. This has been attributed to impaired metabolism and increased nitric oxide, that can reduce synaptic transmission through decreased glutamate release [74] and the accumulation of reactive oxygen species [75]. The contrasting trajectories of hyperthermia's impact in normoxia and hypoxia suggest a nuanced interplay between temperature, synaptic processes, and excitability within unique oxygen environments.

## Hypothermia as potential therapeutic approach

Previous hypothermia studies in comatose post-cardiac arrest patients demonstrated an overall lack of efficacy [1,2] and effects in ischemic stroke patients remain uncertain [15], while fever has been consistently associated with poorer recovery [8–11]. Animal studies, however, consistently demonstrate that hypothermia treatment during experimentally induced ischemic

stroke can reduce infarct volume and improve motor functions [3,4]. An important characteristic of all previous clinical studies is that hypothermia was applied in the recovery phase, mostly after restoration of perfusion [76]. In our study, we demonstrate that hypothermia enhances functional recovery in human neuronal networks, when applied during hypoxia. Pathophysiological mechanisms differ during hypoxia and after reperfusion, with mainly metabolic stress during hypoxia and a myriad of pathomechanisms - including inflammatory responses - during reperfusion [77]. Our results implicate that hypothermia holds potential to provide neuroprotection, when applied in the time window before reperfusion. In particular, hypothermia may be beneficial in ischemic stroke patients to bridge the time between arterial occlusion and revascularization, by slowing down the transition of reversible to irreversible neuronal injury [76].

## Conclusion

Our study demonstrates that hypothermia, applied during hypoxia, significantly enhances post-hypoxic functional recovery in neuronal networks, likely through the preservation of synaptic integrity and improved cell viability. The improvement in neuronal function, synaptic puncta and survival underscores hypothermia's potential as an effective neuroprotective strategy. These findings suggest that hypothermia could play a critical role in preserving brain function during the vulnerable period between arterial occlusion and recanalization in ischemic stroke patients, offering a promising therapeutic approach to mitigate irreversible neuronal damage.

## Supporting information

**S1 Table. Elaborate statistical details of Fig 1.** Statistical analysis relative to Fig 1. Statistical analysis were performed Two-Way ANOVA with Tukey's multiple comparisons test. All comparisons with a p-value < 0.05 are shown.
(DOCX)

**S2 Table. Elaborate statistical details of Fig 2.** Statistical analysis relative to Fig 2. Statistical analysis were performed Two-Way ANOVA with Tukey's multiple comparisons test. All comparisons with a p-value < 0.05 are shown.
(DOCX)

**S3 Table. Elaborate statistical details of Fig 3.** Statistical analysis relative to Fig 3. Statistical analysis were performed Two-Way ANOVA with Tukey's multiple comparisons test for panels b and c, and Sidak's multiple comparisons test for panels e and f. All comparisons with a p-value < 0.05 are shown.
(DOCX)

**S4 Table. Elaborate statistical details of Fig 4.** Statistical analysis relative to Fig 4. Statistical analysis were performed Two-Way ANOVA with Tukey's multiple comparisons test. All comparisons with a p-value < 0.05 are shown.
(DOCX)

**S5 Table. Elaborate statistical details of Fig 5.** Statistical analysis relative to Fig 5. Statistical analysis were performed one-way ANOVA with Dunnet's multiple comparisons test. All comparisons with a p-value < 0.05 are shown.
(DOCX)

## Author Contributions

**Conceptualization:** Eva J. H. F. Voogd, Jeannette Hofmeijer, Monica Frega.

**Data curation:** Eva J. H. F. Voogd.

**Formal analysis:** Eva J. H. F. Voogd.

**Investigation:** Eva J. H. F. Voogd, Marloes Thijs, Marloes R. Levers.

**Methodology:** Eva J. H. F. Voogd, Marloes Thijs, Marloes R. Levers.

**Resources:** Marloes R. Levers.

**Supervision:** Jeannette Hofmeijer, Monica Frega.

**Visualization:** Eva J. H. F. Voogd.

**Writing – original draft:** Eva J. H. F. Voogd, Jeannette Hofmeijer, Monica Frega.

**Writing – review & editing:** Eva J. H. F. Voogd, Marloes Thijs, Marloes R. Levers, Jeannette Hofmeijer, Monica Frega.

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
