## [Decision Letter · Decision Letter 0]

21 Jun 2024

PONE-D-24-12582Hypothermia improves neuronal network recovery in a human-derived in vitro model of the ischemic penumbraPLOS ONE

Dear Dr. Voogd,

Thank you for submitting your manuscript to PLOS ONE. After careful consideration, we feel that it has merit but does not fully meet PLOS ONE’s publication criteria as it currently stands. Therefore, we invite you to submit a revised version of the manuscript that addresses all the points raised during the review process.

Four reviewers evaluated the manuscript. They found it interesting, but all of them asked for further improvements. I do not think that extensive expriments are needed, but additional data were asked that can strengthen the paper, like viability test(s) or additional time-point in the experiments. Please carefully revise the text and make it more organized and clear.==============================

We look forward to receiving your revised manuscript.

Kind regards,

Mária A. Deli, M.D., Ph.D.

Academic Editor

PLOS ONE

 [This research has been supported by an institutional research grant.].  

5. In the online submission form, you indicated that [Data cannot be shared publicly because of the size of the data. The data that support the findings of this study are available upon reasonable request from the authors.]. 

Reviewers' comments:

Reviewer's Responses to Questions

**Comments to the Author**

1. Is the manuscript technically sound, and do the data support the conclusions?

Reviewer #1: Partly

Reviewer #2: Yes

Reviewer #3: Partly

Reviewer #4: Partly

2. Has the statistical analysis been performed appropriately and rigorously? 

Reviewer #1: Yes

Reviewer #2: Yes

Reviewer #3: Yes

Reviewer #4: Yes

3. Have the authors made all data underlying the findings in their manuscript fully available?

Reviewer #1: Yes

Reviewer #2: Yes

Reviewer #3: Yes

Reviewer #4: No

4. Is the manuscript presented in an intelligible fashion and written in standard English?

Reviewer #1: Yes

Reviewer #2: Yes

Reviewer #3: No

Reviewer #4: Yes

5. Review Comments to the Author

Reviewer #1: The manuscript of Voogd et al. presents intriguing data, further confirming the previous finding that hypothermia applied during hypoxia may have neuroprotective effects. The topic is timely and of significant importance, as the prevalence of ischemic stroke is continuously increasing worldwide, however no neuroprotective agents have been demonstrated to impact clinical outcomes.

The authors monitored the electrophysiological activity throughout the hypoxic treatment until the end of the recovery period, thereby providing a complete, comprehensive picture of how temperature affects neuronal activity. However a similar comprehensive tracking of the number of synaptic puncta is missing from the study. In my opinion, it would be highly informative to investigate the number of synaptic puncta at the end of the recovery period as well. Furthermore, there is no information available regarding the viability of cells. Authors mentioned in the discussion, that under hypoxia, hyperthermia may induce apoptosis. Is it possible, that the loss of neuronal network functioning observed in this this study may be due to cell death?

Reviewer #2: The authors established a neuronal network model from hiPSC for MEA neuronal activity test in their previous publication. In this study, they used this model to investigate the effects of hypothermia and hyperthermia on neuronal activity after hypoxia.

1. In the title, “in vitro model of the ischemic penumbra” is incorrect. In the penumbra, blood flow is typically reduced to 30%-70%. The authors used 2% hypoxia which provides too little oxygen but too much glucose to accurately mimic penumbra situation in vitro. A modified oxygen-glucose deprivation (OGD) model with 5-8% hypoxia and 30% glucose content is recommended.

2. MEA technology has been utilized for over two decades, allowing numerous researchers to study neuronal activity under various physiological and pathological conditions, such as hypothermia, hyperthermia, and hypoxic stress, using either rodent, sheep or human neurons (e.g., PMID: 17081617; PMID: 17093117; PMID: 34975426; PMID: 32320794). It would be informative if the authors used patient-derived neurons, despite the difficulty in obtaining them from ischemic patients. Or, alternatively they may at least use primary human neurons from other diseases. iPSC-derived neurons remain quite artificial, and their results do not provide significant new insights in this field. It is already known that hypothermia is neuroprotective and promotes the recovery of neuronal activity after hypoxia, but it has the opposite effects under normal conditions. Conversely, hyperthermia is known to exacerbate neuronal damage after hypoxia.

3. To enhance the novelty of the present study, I recommend that the authors test the effects of different temperatures and oxygen levels in their model. The author noted that in some reports, even 36°C is neuroprotective after hypoxia. Providing a systematic view of temperature effects could be valuable for therapeutic applications.

Reviewer #3: About the manuscript PONE-D-24-12582, entitled ”Hypothermia improves neuronal network recovery in a human-derived in vitro model of the ischemic penumbra”, by Franca EJH et al., submitted to PLOS ONE.

While I consider it interesting, the manuscript is rather difficult to read. I would suggest to improve the whole presentation, clarifying the presentation of the results and very much the discussion.

General

The manuscript reports experiments with neuronal networks derived from human induced pluripotent stem cells, monitored with micro-electrode arrays, to investigate how treatment with normothermia, hypothermia and hyperthermia during hypoxia influences neuronal functionality, improving post-hypoxic recovery. The neuronal networks are also evaluated with immunocytochemistry, studying the effect of the treatment on synaptic puncta.

Methods

The description of the methods is detailed, but not well organised, including experimental steps that are not considered when describing the results. The authors describe the protocol for identifying excitatory and inhibitory neurons, but the phenotype is not confirmed nor used for evaluating the effects of hypoxia and/or temperature conditions.

Results

While the experimental timing is schematically described in Fig. 1, 2 and 4, it is not clear enough when the recording is performed. Is it continuously performed, as indicated in Fig.1e-g? But, when are the samples taken for building the graphics at Fig. 1i-k? Please, clarify that for all figures, completing and making it uniform along the figure legends.

Would one understand that the sampling in Fig. 1d, Fig.2b and Fig.4b was taken at the beginning or at the end of a period of 6h, 24h or 48h of asphyxia? Under continuous either normothermia, hypothermia or hyperthermia?, but a final 6h normoxia/normothermia period was also performed, at least for Fig. 4? Please, uniform the layout of the figures. In Fig. 1i-k; Fig. 2g-I and Fig. 4i-l, why not using the same scales? The dotted line represents 100% or 1? Also, in Fig. 4k?

The timing for Fig. 3 has to be clarified, making possible for the reader to rapidly understand when the tissue was fixed. After 6h, 24h and 48h of hypoxia? The labelling of synapsin ½ is not evident, at least in the figure I received. The scale bar in Fig. 3a is hardly seeing.

Before evaluating the temperature variable, please, describe clearly on the effect of hypoxia, I guess Control versus Normothermia, describing the time course. Clarify, please, the issue of the effect of 48h of hypoxia, inducing only 50% of decrease of synaptic puncta?

Is that a severe or mild effect? How should be that when comparing with in vivo models? Any specific effect on excitatory or inhibitory neurons, or neurons versus glial cells?

Discussion

I would recommend to start by discussing the relevance and advantages of the experimental mode, also summarising the obtained results, making clear what are we learning with the results. Also, please, explain about the relevance of the effects observed after hypothermia, producing outcomes surpassing the control condition?

How long does it take for seeing a deleterious effect of hyperthermia? 48h (as seeing in Fig.3b-d??

Discuss please about the physiological relevance of the obtained results. What is the relevance of an increase in synaptic puncta following 48h hypoxia + hypothermia?

In general, the discussion is floppy, referring to issues that are not considered by the selected experimental protocol. I would suggest to concentrate on the observed results, expanding then on their relevance, physiologically and/or clinically.

The discussion in lines 382-396 should be improved, expanding the idea that hypothermia would reinforce inhibitory synapses? How is that? considering what is shown in Fig. 3 and 4? What’s about effects on excitatory and/or inhibitory neurons?

How to deal with the discussion expressed in line 406-412 and the effects shown in Fig. 2?

I would suggest that the authors focus on what was observed in the study.

The issue of a therapeutic strategy before a clinical outcome could perhaps deserve some commentaries.

The conclusion should be more convincing. Hypothermia should prevent synaptic loss or rather increase functional efficacy?

The final sentence “freezing the penumbra” is justifying the title? Indeed, the issue of penumbra is just taken at the end of the article. How should the chosen experimental approach be a model for describing what is occurring in the penumbra? Is the issue of “penumbra” justified for being inserted in the title?

Reviewer #4: Review of “Hypothermia improves neural network recovery…”

This manuscript evaluates the effects of hypothermic (34C) and hyperthermic (39C) conditions on human-derived neuronal network plated on microelectrode arrays during the application of hypoxia. Functional readouts are performed prior, during and after application of 48hrs of hypoxic conditions at normothermic, hypothermic and hyperthermic temperatures. The results show that the application of hypothermia had some beneficial effects on the network’s electrophysiology, specifically the firing and burst rate whereas the application of hyperthermia had a generally significantly worse outcome than normothermia.

The manuscript is generally well-written and organized, and the data presented is by en large clear. The study is somewhat limited in its focus (primarily focusing on electrical activity and 2 antibody stains), and I have a few comments I’d like the authors to address before I recommend this manuscript to be published:

1. While the data in Fig. 1 shows that there is a clear benefit of applying hypothermic (34C) conditions to the neuronal networks during under hypoxic conditions - it is unclear how viable the culture is 24hrs or even 48hrs after the application of hypothermia. It would significantly strengthen the paper if the authors could provide data further out then just 6hrs (see for example Scimone et al. PlosOne, 2020 - application of hypothermia to TBI).

2. The statement on Line 427 that hypothermia enhances functional recovery during hypoxia might be misleading if the long-term outcome of the culture is unaltered, i.e., if the long-term (24-48hr) outcome does not show any statistical significant effect from normothermia under hypoxic conditions.

3. It would be very helpful for all figures showing time on the x-axis (e.g., Fig.1 e-g, etc) to show the full time data (10min initially, 48 hrs + 6 hrs post) and delineate the different regimes (with shaded regions for example) clearly on the plots. Please also add raster plots for the 6hrs post hypoxia for all groups.

4. It would be great to have some general viability data alongside the electrophysiological data to assess the general health of the neural cell populations in particular at the final time point.

5. Line 442 in the conclusion “freezing the penumbra” is somewhat misleading as we are talking about a small change (37C to 34C) in temperature (not 0C), which would have significant adverse effects.

6. The fact that hypothermia had such a strong effect on the control samples is somewhat concerning. Some loss in cell function (and most likely cell viability) is certainly expected (this is also shown in the Scimone et al as an in vitro example) but coming back to my point 1., without showing the long term, stable benefit of the application of hypothermia the overall study impact might be significantly diminished. Also, I was somewhat surprised to see, on the flip side, the almost positive effect of hyperthermia on the control cases - some more elaboration on what the authors think might be happening here within the context of theirs and other, similar in vitro culture systems will be useful.

6. PLOS authors have the option to publish the peer review history of their article (what does this mean?). If published, this will include your full peer review and any attached files.

Reviewer #1: No

Reviewer #2: No

Reviewer #3: No

Reviewer #4: No

---

## [Author Response · Author response to Decision Letter 0]

15 Oct 2024

Reviewer #1: 

The manuscript of Voogd et al. presents intriguing data, further confirming the previous finding that hypothermia applied during hypoxia may have neuroprotective effects. The topic is timely and of significant importance, as the prevalence of ischemic stroke is continuously increasing worldwide, however no neuroprotective agents have been demonstrated to impact clinical outcomes.

The authors monitored the electrophysiological activity throughout the hypoxic treatment until the end of the recovery period, thereby providing a complete, comprehensive picture of how temperature affects neuronal activity. However a similar comprehensive tracking of the number of synaptic puncta is missing from the study. In my opinion, it would be highly informative to investigate the number of synaptic puncta at the end of the recovery period as well. Furthermore, there is no information available regarding the viability of cells. Authors mentioned in the discussion, that under hypoxia, hyperthermia may induce apoptosis. Is it possible, that the loss of neuronal network functioning observed in this this study may be due to cell death?

Synaptic puncta. We thank the reviewer for his/her comments and we agree that the extension of the time points of synaptic puncta is highly informative. Therefore we added data on synaptic puncta at 6 and 24 hours after recovery for normothermic and hypothermic neuronal networks. We found at 24 hours of recovery a significantly higher number of synaptic puncta in hypothermic neuronal networks as compared to normothermic. We added these additional results to Figure 4 (e-g) and the following text:

Line 357-378: “Next, we examined the recovery of neuronal networks under normothermic and hypothermic conditions, excluding hyperthermic networks because of the absence of functional recovery after hypoxia (fig. 4 e-g). After 6 hours of recovery, normothermic and hypothermic neuronal networks had 0.84 ± 0.07 synaptic puncta per 10 μm and 0.99 ± 0.08 synaptic puncta per 10 μm , respectively (p > 0.9999). After 24 hours of recovery, hypothermic neuronal networks had a higher number of synaptic puncta per 10 μm than normothermic neuronal networks (1.6 ± 0.11 vs. 0.8 ± 0.09; p<0.0001).

Fig. 4. Number of synaptic puncta increased during and after hypoxia in hypothermic neuronal networks. e. Representative images showing synaptic puncta (arrowheads, positive for synapsin 1/2; red) along the dendrite (positive for MAP2; green; nuclei positive for DAPI; blue) in normothermic and hypothermic neuronal networks after 6 and 24 hours of recovery. Scale bar: 10 μm. f-g. Bar graphs showing the number of synaptic puncta per 10 μm dendrite in neurons exposed to normothermia (n=4) or hypothermia (n=6) after f. 6 hours and g. 24 hours of recovery. Black line represents baseline number of synaptic puncta (n = 11). *P < 0.05, **P < 0.005, ***P < 0.0005, ****P < 0.0001. Two-way ANOVA with Tukey’s multiple comparison analyses were performed between conditions. Exact p values are reported in table S3. “

Live, dead and apoptotic cells. We agree with the reviewer that the viability of the neuronal networks during the different temperature regimens and hypoxia holds valuable information. Therefore, we added data on cell viability including apoptosis after 24 and 48 hours of hypoxia for normothermic, hypothermic and hyperthermic neuronal networks, as well as 6 and 24 hours of recovery for normothermic and hypothermic neuronal networks. We found that, after 24 hours of hypoxia, neuronal networks treated with hypothermia had significantly fewer dead cells compared to normothermia. Hyperthermic networks showed similar cell viability as normothermia. After 48 hours of hypoxia, neuronal networks treated with hypothermia showed no significant differences in cell viability compared to normothermic networks. Hyperthermic networks showed significantly less live cells and more dead cells compared to normothermic networks. During recovery, we found after 6 hours, normothermic networks had significantly more live cells compared to hypothermic networks. However, after 24 hours of recovery we found that hypothermic networks had significantly better outcomes, by showing more live cells, and less apoptotic cells.

The results are reported in a new figure (new Figure 3). We added the following paragraph to the results, and the complete description of the methodology for the identification of live, dead and apoptotic cells in the Material and Methods section (line163-178).

Line 307-341: “Hypothermia improves cell viability after hypoxia

We next investigated the effect of normo-, hypo-, and hyperthermia on cell viability (Fig. 3a-b). After 24 hours of hypoxia, under normothermia we observed 36% live cells, 11% apoptotic cells, and 52% dead cells. Under hypothermia, there was a slightly higher percentage of live cells (48%) and apoptotic cells (19%), with significantly fewer dead cells (34%) compared to normothermia (p = 0.0006). Hyperthermia resulted in 34% live cells, 10% apoptotic cells, and 56% dead cells, showing a similar pattern to normothermia with no significant differences in live, apoptotic or dead cell counts.

After 48 hours of hypoxia, hypothermic networks maintained a higher percentage of live cells (61%) compared to normothermic networks (55%), and fewer dead cells (22% vs. 32%), although this difference was not statistically significant. Hyperthermia was associated with a significant decline in cell viability, exhibiting the lowest percentage of live cells (33%) and the highest percentage of dead cells (63%). Significant differences were found between normothermic and hyperthermic networks in both live cell count (p = 0.0129) and dead cell count (p = 0.0001).

Next, we examined the recovery of neuronal networks under normothermic and hypothermic conditions, excluding hyperthermic networks because of the absence of functional recovery after hypoxia (fig. 3d-f). After 6 hours of recovery, under normothermia, neuronal networks showed a higher percentage of live cells (62%) compared to hypothermic networks, although not statistically significant. However, after 24 hours of recovery, hypothermic neuronal networks demonstrated significantly better outcomes, with 93% live cells compared to 37% in normothermic networks (p < 0.0001). Additionally, hypothermic networks had significantly fewer apoptotic cells (16%) compared to normothermic networks (49%; p < 0.0001).”

Fig. 3 Hypothermia improves cell viability after hypoxia. a. Representative images showing cell viability including apoptosis of neuronal networks exposed to 24 or 48 hours hypoxia and normothermia, hypothermia or hyperthermia. Neuronal networks stained for DAPI (blue), Cell event Caspase-3/7 (green) and propidium iodine (red). Scale bar 30 μm. b-c. Stacked bar graphs showing the percentage of live, apoptotic (apo) and dead cells after b. 24 hours of hypoxia and c. 48 hours of hypoxia for normothermia (n=9), hypothermia (n=7) and hyperthermia (n=3). d. Representative images showing cell viability including apoptosis of neuronal networks exposed to 6 or 24 hours of recovery after hypoxia and normothermia or hypothermia. Neuronal networks were stained for DAPI (blue), Cell event Caspase-3/7 (green) and propidium iodine (red). e-f. Stacked bar graphs showing the percentage of live, apoptotic (apo) and dead cells after e. 6 hours of recovery or f. 24 hours of recovery for normothermia (n=4) and hypothermia (n=6). *P < 0.05, **P < 0.005, ***P < 0.0005, ****P < 0.0001. Two-way ANOVA with b-c. Tukey’s multiple comparisons and e-f. Sidak’s multiple comparisons test. “

Reviewer #2: 

The authors established a neuronal network model from hiPSC for MEA neuronal activity test in their previous publication. In this study, they used this model to investigate the effects of hypothermia and hyperthermia on neuronal activity after hypoxia.

1. In the title, “in vitro model of the ischemic penumbra” is incorrect. In the penumbra, blood flow is typically reduced to 30%-70%. The authors used 2% hypoxia which provides too little oxygen but too much glucose to accurately mimic penumbra situation in vitro. A modified oxygen-glucose deprivation (OGD) model with 5-8% hypoxia and 30% glucose content is recommended.

We thank the reviewer for his / her comment. We realize that the term "penumbra" was not directly addressed until the end of the discussion. Our model mimics oxygen deprivation in general, which is a key feature of the ischemic penumbra. To better reflect the scope of our work, we adjusted the title to: “Hypothermia improves neuronal network recovery in a human-derived in vitro model of oxygen-deprivation”. 

We are aware that the level of oxygen used in this study (2% O2) is not in line with what is observed in vivo. This type of oxygen-deprivation is essential for replicating the conditions under which neuronal networks become electrically silent over time, yet remaining potentially salvageable upon restoration of normoxia, which is a key characteristic of the ischemic penumbra. With higher oxygen levels, the neuronal networks remain functionally active (as also shown in a rodent in vitro model [1]), which hampers testing of neuroprotective investigations. While glucose was not lowered in our initial setup, our previous studies have shown that long-term hypoxia results in decreased glucose levels by consumption [2, 3], aligning it with in vivo penumbral conditions, where glucose is also reduced by consumption. Thus, we believe that our results can be relevant for ischemic stroke research.

2. MEA technology has been utilized for over two decades, allowing numerous researchers to study neuronal activity under various physiological and pathological conditions, such as hypothermia, hyperthermia, and hypoxic stress, using either rodent, sheep or human neurons (e.g., PMID: 17081617; PMID: 17093117; PMID: 34975426; PMID: 32320794). It would be informative if the authors used patient-derived neurons, despite the difficulty in obtaining them from ischemic patients. Or, alternatively they may at least use primary human neurons from other diseases. iPSC-derived neurons remain quite artificial, and their results do not provide significant new insights in this field. It is already known that hypothermia is neuroprotective and promotes the recovery of neuronal activity after hypoxia, but it has the opposite effects under normal conditions. Conversely, hyperthermia is known to exacerbate neuronal damage after hypoxia.

We acknowledge that the use of microelectrode array (MEA) technology has indeed been extensive and highly informative in studying neuronal activity under various conditions. However, the suggestion to use patient-derived neurons or primary human neurons from other diseases is, while valuable, beyond the scope of our current study for several reasons. Using patient-derived neurons, especially from stroke patients, would require a significant shift in resources, methodology, and scope, as well as approval of ethical committees. It would be a whole new study in terms of research objectives and experimental design, which is beyond the boundaries of our current study's aims. We agree that patient-derived neurons and primary human neurons are valuable models for translational research. However, we believe that using well-characterized healthy induced pluripotent stem cell (iPSC)-derived neurons provides a valuable first step, by providing a controlled and reproducible system to investigate the effects of hypoxia and temperature regimens [4, 5]. These cells allow us to focus on the electrophysiological responses to hypoxia and hypothermia/hyperthermia without the additional complexities introduced by disease-specific or patient-specific variations, thereby allowing us to contribute meaningful and reproducible data to the field.

3. To enhance the novelty of the present study, I recommend that the authors test the effects of different temperatures and oxygen levels in their model. The author noted that in some reports, even 36°C is neuroprotective after hypoxia. Providing a systematic view of temperature effects could be valuable for therapeutic applications.

We appreciate the reviewer's suggestion to test the effects of different temperatures and oxygen levels in our model. The effect of different oxygen levels has already been studied in detail [1]. Based on those findings, we selected the oxygen level that strikes a balance between inducing a significant loss of neuronal functionality and providing a window for evaluating neuroprotective strategies. This level allows us to model conditions where neuronal networks are impaired but not irreversibly damaged, ensuring that interventions such as hypothermia can still offer protective effects, rather than inducing widespread cell death. 

It would be interesting to investigate a broader range of temperatures, but this would require a significant number of additional experiments. That goes beyond the current scope of our study. Our primary goal was to determine whether hypothermia could offer neuroprotection and whether hyperthermia would exacerbate neuronal damage in our in vitro model.

The temperatures we selected —34°C for hypothermia and 39°C for hyperthermia— were based on values reported in the literature that are clinically relevant. We already observe significant protective effects at 34°C, and further temperature testing, although valuable in its own right, would not necessarily provide novel insights for the specific aims of our study. Furthermore, while we see a clear neuroprotective effect at hypothermia at 34°C, these findings are not intended to prescribe a precise temperature range for clinical application, since such a translation from in vitro to in vivo is challenging. The main takeaway for therapeutic purposes is that hypothermia is neuroprotective by synapse recovery and improved cell viability and not that a specific temperature should be used.

Given these considerations, we believe the temperatures we chose effectively demonstrate the protective role of hypothermia and the deleterious role of hyperthermia. Further testing is outside the intended focus of this research.

Reviewer #3: 

About the manuscript PONE-D-24-12582, entitled ”Hypothermia improves neuronal network recovery in a human-derived in vitro model of the ischemic penumbra”, by Eva JHF et al., submitted to PLOS ONE.

While I consider it interesting, the manuscript is rather difficult to read. I would suggest to improve the whole presentation, clarifying the presentation of the results and very much the discussion.

We thank the reviewer for considering our work interesting. We used the reviewer’s suggestions to improve the readability of the manuscript. See responses below for a more detailed explanation of these improvements.

General

The manuscript reports experiments with neuronal networks derived from human induced pluripotent stem cells, monitored with micro-electrode arrays, to investigate how treatment with normothermia, hypothermia and hyperthermia during hypoxia influences neuronal functionality, improving post-hypoxic recovery. The neuronal networks are also evaluated with immunocytochemistry, studying the effect of the treatment on synaptic puncta.

Methods

The description of the methods is detailed, but not well organised, including experimental steps that are not considered when describing the results. The authors describe the protocol for identifying excitatory and inhibitory neurons, but the phenotype is not confirmed nor used for evaluating the effects of hypoxia and/or temperature conditions.

We re-organized the method section to make it clearer and more in line with the presentation of the results. Textual changes are in red in the manuscript (lines 69-222). 

We apologize for the confusion in our Material and Methods section. We have not described the methodology for identifying excitatory and inhibitory neurons, but the protocol to differentiate such cell types from hiPSCs. In this study, we indeed included both excitatory and inhibitory neurons because their combined activity significantly influences overall neuronal functionality, providing a more physiologically relevant representation of neuronal populations in the brain, though not identical to in vivo co

---

## [Decision Letter · Decision Letter 1]

19 Nov 2024

Hypothermia improves neuronal network recovery in a human-derived in vitro model of oxygen-deprivation

PONE-D-24-12582R1

Dear Dr. Voogd,

We’re pleased to inform you that your manuscript has been judged scientifically suitable for publication and will be formally accepted for publication once it meets all outstanding technical requirements.

Kind regards,

Mária A. Deli, M.D., Ph.D.

Academic Editor

PLOS ONE

Additional Editor Comments (optional):

Reviewers' comments:

Reviewer's Responses to Questions

**Comments to the Author**

1. If the authors have adequately addressed your comments raised in a previous round of review and you feel that this manuscript is now acceptable for publication, you may indicate that here to bypass the “Comments to the Author” section, enter your conflict of interest statement in the “Confidential to Editor” section, and submit your "Accept" recommendation.

Reviewer #1: All comments have been addressed

Reviewer #4: All comments have been addressed

2. Is the manuscript technically sound, and do the data support the conclusions?

Reviewer #1: Yes

Reviewer #4: Yes

3. Has the statistical analysis been performed appropriately and rigorously? 

Reviewer #1: Yes

Reviewer #4: Yes

4. Have the authors made all data underlying the findings in their manuscript fully available?

Reviewer #1: Yes

Reviewer #4: Yes

5. Is the manuscript presented in an intelligible fashion and written in standard English?

Reviewer #1: Yes

Reviewer #4: Yes

6. Review Comments to the Author

Reviewer #1: (No Response)

Reviewer #4: (No Response)

7. PLOS authors have the option to publish the peer review history of their article (what does this mean?). If published, this will include your full peer review and any attached files.

Reviewer #1: No

Reviewer #4: No

---

## [Editor Report · Acceptance letter]

11 Dec 2024

PONE-D-24-12582R1 

PLOS ONE

Dear Dr. Voogd, 

I'm pleased to inform you that your manuscript has been deemed suitable for publication in PLOS ONE. Congratulations! Your manuscript is now being handed over to our production team.

Kind regards, 

on behalf of

Prof. Mária A. Deli 

Academic Editor

PLOS ONE